# Dynamic Mechanical Control of Alginate-Fibronectin Hydrogels with Dual Crosslinking: Covalent and Ionic

**DOI:** 10.3390/polym13030433

**Published:** 2021-01-29

**Authors:** Sara Trujillo, Melanie Seow, Aline Lueckgen, Manuel Salmeron-Sanchez, Amaia Cipitria

**Affiliations:** 1Centre for the Cellular Microenvironment, University of Glasgow, 72-76 Oakfield Avenue, Glasgow G12 8LT, UK; Sara.TrujilloMunoz@glasgow.ac.uk (S.T.); mseow99@gmail.com (M.S.); 2Centre for Biomaterials and Tissue Engineering (CBIT), Universitat Politècnica de València, Camino de Vera s/n, 46022 Valencia, Spain; 3Julius Wolff Institute & Centre for Musculoskeletal Surgery, Charité—Universitätsmedizin Berlin, 13353 Berlin, Germany; aline.lueckgen@charite.de; 4Biomedical Research Networking Centre in Bioengineering, Biomaterials and Nanomedicine (CIBER-BBN), 46022 Valencia, Spain; 5Max Planck Institute of Colloids and Interfaces, Department of Biomaterials, 14476 Potsdam, Germany

**Keywords:** alginate hydrogel, fibronectin, enzymatic degradation, dual crosslinking, covalent and ionic crosslinking, dynamic mechanical properties

## Abstract

Alginate is a polysaccharide used extensively in biomedical applications due to its biocompatibility and suitability for hydrogel fabrication using mild reaction chemistries. Though alginate has commonly been crosslinked using divalent cations, covalent crosslinking chemistries have also been developed. Hydrogels with tuneable mechanical properties are required for many biomedical applications to mimic the stiffness of different tissues. Here, we present a strategy to engineer alginate hydrogels with tuneable mechanical properties by covalent crosslinking of a norbornene-modified alginate using ultraviolet (UV)-initiated thiol-ene chemistry. We also demonstrate that the system can be functionalised with cues such as full-length fibronectin and protease-degradable sequences. Finally, we take advantage of alginate’s ability to be crosslinked covalently and ionically to design dual crosslinked constructs enabling dynamic control of mechanical properties, with gels that undergo cycles of stiffening–softening by adding and quenching calcium cations. Overall, we present a versatile hydrogel with tuneable and dynamic mechanical properties, and incorporate cell-interactive features such as cell-mediated protease-induced degradability and full-length proteins, which may find applications in a variety of biomedical contexts.

## 1. Introduction

The extracellular matrix (ECM) is a dynamic, water-saturated scaffold that undergoes remodelling to maintain homeostasis during development, disease or injury [1]. Throughout these processes, there are many biophysical and biochemical cues that can trigger changes in cell behaviour including bulk ECM mechanics [2]. For instance, matrix mechanical properties have been shown to dictate stem cell differentiation, where stiffer matrices promote osteogenesis, whereas softer matrices trigger adipogenesis in mesenchymal stem cells, in both 2D and 3D scenarios [3,4,5]. Stem cells are also able to retain memory of their mechanical history, i.e., the stiffness of substrates on which they have been grown, which determines their commitment to different lineages [6]. Furthermore, stiff matrices have been shown to induce fibroblast cells to switch to a profibrotic behaviour [7]. There have been advances in materials science to investigate the effects of matrix mechanics on cell phenotypes. Still, there remains a need to develop biomaterials that recapitulate the dynamic mechanical [8,9,10,11,12,13,14] and biological aspects of the ECM, to model the evolution of complex pathological processes such as fibrosis [14,15,16,17]. In addition to playing a role in cell differentiation and tissue pathologies, matrix mechanics are also important in applications such as drug delivery and wound healing [18,19,20,21].

Alginate is a polysaccharide derived from seaweed composed of α-d-mannuronic acid (M residue) and β-l-guluronic acid (G residue). In particular, alginate can be found as different linear copolymers containing blocks of M and G residues [21]. These blocks are composed of either repeated G residues (GGGGGG), sequential M residues (MMMMMM) or alternating M/G residues (GMGMGM). M and G contents in alginate depend on the extraction source, while only G-blocks are thought to interact with divalent cations (e.g., Ca^2+^) to form ionically crosslinked hydrogels. Therefore, the M/G ratio of alginate is critical for the final physical properties of the fabricated hydrogel [21,22,23].

The most widely used method to fabricate alginate hydrogels is to combine the alginate solution with divalent cations, most commonly calcium chloride (CaCl_2_). The mild gelation conditions with divalent cations such as Ca^2+^ or Mg^2+^ that enable the simultaneous incorporation of cells, is one of the reasons why alginate has been classically used for biomedical applications such as drug delivery [24], wound dressings [25] and tissue engineering [26,27]. This material is also biocompatible, has low toxicity and relatively low cost [21,28]. Ionic crosslinking typically occurs via fast and weakly controlled gelation due to the high solubility of calcium chloride in aqueous solutions. One significant disadvantage of ionically crosslinked alginate hydrogels is their limited stability in physiological conditions, as divalent cations are released in aqueous media and hydrogels tend to dissolve over time in an uncontrolled manner. This shortcoming has stimulated interest in the development of covalently crosslinked alginate hydrogel strategies [29,30,31]. To this end, alginate has been modified to present different functional groups such as norbornene moieties—resulting in alginate chains that can then be crosslinked with dithiol crosslinkers via ultraviolet (UV) light-initiated thiol-ene chemistry [32]. This strategy has also been used with other polymers such as hyaluronic acid (HA) or poly(ethylene) glycol (PEG), demonstrating that thiol-ene chemistry is cytocompatible [33,34,35,36].

Alginate lacks biologically active sites that can interact with cells. This is an interesting aspect of alginate as it can be used as a base scaffold that can be further engineered to incorporate defined biological functionalities that allow controlled interactions with cells. For instance, alginate can be decorated with the Arg-Gly-Asp sequence (RGD peptide), an extensively used cell adhesion cue, to allow cells to attach and spread [22,37,38]. Additionally, alginate hydrogels have been formed using enzymatically degradable crosslinkers so cells can remodel the biomaterial by secreting specific proteases [29]. This allows cells to migrate and infiltrate the material, a requirement for many tissue engineering applications [29,39,40,41].

Fibronectin (FN) is an ECM protein that contains the cell-adhesive RGD peptide among other cues [42,43,44,45]. As it triggers cells adhesion, FN has been extensively used to coat plates and facilitate cell culture. Furthermore, FN also binds numerous growth factors in close proximity to the cell-adhesion domain which enhances growth factor downstream signalling [46,47,48]. This feature has been used to engineer FN-based hydrogels that allow synergistic integrin-growth factor signalling [49,50,51]—i.e., incorporating FN into 3D hydrogels enhances the biological activity of the system by promoting cell adhesion and enabling growth factor signalling.

Here, we take advantage of the ability of norbornene-modified alginate to be crosslinked both ionically and covalently to fabricate dual crosslinked hydrogels. We confirm control over the system’s mechanical properties. In addition, we use thiol-ene chemistry to functionalise alginate with FN and incorporate protease-sensitive crosslinkers. We demonstrate that the hydrogel’s mechanical properties can be tuned dynamically by adding or removing calcium ions from the medium. Such a versatile hydrogel system offers multiple opportunities for biomedical applications such as drug delivery or tissue engineering.

## 2. Materials and Methods

### 2.1. Alginate Modification

Alginate was modified following a previously published procedure [29,30]. Briefly, high molecular weight (265 kDa), high guluronic acid, sodium alginate (Pronova UP MVG; NovaMatrix, Sandvika, Norway) was dissolved at room temperature in 2-(*N*-morpholino)ethanesulfonic acid (MES, 0.1 M, Merck, Eppelheim, Germany) and NaCl (0.3 M, Millipore, Eppelheim, Germany) buffer at pH 6.5 for 16 h. Then, *N*-hydroxysuccinimide (NHS; Sigma-Aldrich, Eppelheim, Germany) and 1-ethyl-3-(3-dimethylaminopropyl)-carbodiimide hydrochloride (EDC; Sigma-Aldrich, Eppelheim, Germany) were added dropwise at a molar ratio 5000:1 to the alginate solution. After that, 5-norbornene-2-methylamine (TCI Deutschland GmbH, Eschborn, Germany) was added to introduce norbornene moieties to the alginate backbone at a theoretical degree of substitution of 200 molecules per alginate chain. This reaction was run at room temperature for 20 h and stirring at 700 rpm. After 20 h, the reaction was quenched by adding hydroxylamine (Sigma-Aldrich, Eppelheim, Germany). Dialysis (Spectra/Por 6, MWCO 3.5 kDa; Spectrum, New Brunkswick, NJ, USA) against a salt gradient (6 g·L^−1^ to 0 g·L^−1^; Sigma-Aldrich, Eppelheim, Germany) was performed in deionized water for 3 days with 3 to 4 changes per day. The solution was purified with activated charcoal (Sigma-Aldrich, Eppelheim, Germany), sterile-filtered (0.22 µm; Steriflip-GP; Merck, Eppelheim, Germany), and lyophilized.

### 2.2. Fibronectin PEGylation

Fibronectin (FN, 3 mg·mL^−1^, YoProteins, Ronninge, Sweden) was poly(ethylene) glycol (PEG)ylated following a previously published method [47]. Briefly, FN was dissolved in 5 mM Tris(2-carboxyethyl)phosphine hydrochloride (pH 7, Sigma-Aldrich, Eppelheim, Germany) and 8 M urea (Acros Organics, Geel, Belgium, 99.5%) in phosphate buffer saline (PBS, Gibco, pH 7.4) over 15 min at room temperature. Then, 4-arm-PEG-Maleimide (PEGMAL, 20 kDa, LaysanBio, Arab, AL, USA) was incubated for 30 min at room temperature at a molar ratio 1:4 FN to PEGMAL. The reaction was quenched by alkylation using 14 mM iodoacetamide (Sigma-Aldrich, Eppelheim, Germany) in PBS (pH 8; 2 h). After dialysis (Mini-A-Lyzer, MWCO 10 kDa, ThermoFisher, Eindhoven, The Netherlands), the protein solution was precipitated using cold ethanol (Fisher, Loughborough, UK). The protein pellet was dissolved in 8 M urea, dialysed against PBS and used at a final concentration of 2.5 mg·mL^−1^.

### 2.3. Alginate Hydrogel Fabrication

Norbornene-modified alginate was dissolved in PBS at either 2% or 3% *w*/*v* (2 wt% and 3 wt%). Irgacure 2959 (2-Hydroxy-4′-(2-hydroxyethoxy)-2-methylpropiophenone, Sigma-Aldrich, Eppelheim, Germany) was added to the alginate solution at 0.05% *w*/*v* and the crosslinker was also added to the mixture at different thiol to norbornene molar ratios (crosslinking ratio, X). Two different types of crosslinkers were used—either dithiothreitol (DTT, Sigma-Aldrich, Eppelheim, Germany) or protease-degradable custom synthesised peptide (VPM, GCRDVPMSMRGGDRCG, purity 96.9%, M_w_ 1696.96 Da, GenScript, Leiden, The Netherlands). PEGylated FN was also added to the mixture at a final concentration of 1 mg·mL^−1^ when preparing fibronectin-alginate hydrogels. Hydrogel composition is detailed in Table 1, where the crosslinking ratio X is defined as the ratio of thiol groups to norbornene groups. This mixture was transferred to a PDMS circular mould (16 mm diameter and 1.4 mm thickness for rheology experiments and 6 mm diameter and 1 mm thickness for swelling and release assays) and flattened using a glass slide. Then, the moulds were UV irradiated for 10 min at 10 mW·cm^−2^.

### 2.4. Rheological Measurements

Rheological measurements were carried out with a stress-controlled rheometer (MCR302, Anton Paar Instrument, Graz, Austria) fitted out with a parallel plate geometry (upper plate diameter used: 15 mm) and with controlled temperature set to 25 °C. Frequency sweep tests in the range of 1 to 100 rad·s^−1^ were performed with a normal force of 1 N and a strain of 0.1%, to determine the regime where the shear storage modulus, G’, was independent of frequency, ω. Strain sweeps in the range of 0.01% to 1% and angular frequency of 10 rad·s^−1^ were performed to determine the elastic shear modulus of the material. Samples were measured at swelling equilibrium and hydration was preserved by addition of PBS at the exposed sides of the samples. Results are reported as stiffness (Young’s modulus, *E*) that was obtained from the storage modulus, *G*’, following *E* = 2*G*’ (1 + υ), where υ is the Poisson’s ratio, assumed to be 0.5. All conditions were assessed in triplicate.

### 2.5. Stiffening–Softening Assays

To assess dynamic stiffening–softening of hydrogels, samples were formed by covalent crosslinking using UV-initiated polymerisation as described above. Samples were immersed in PBS and let to swell until equilibrium (24 h), when they were mechanically tested using rheology. For subsequent ionic crosslinking, samples were immersed in 250 mM CaCl_2_ (Sigma-Aldrich, Eppelheim, Germany) for 2 h at room temperature and mechanically assessed again. In order to soften the hydrogels by removing calcium ions, samples were either immersed in PBS or 0.25 M EDTA for 3 days and mechanically tested again. Finally, to evaluate reversible stiffening, the hydrogels were immersed in 250 mM CaCl_2_ again for 2 h and mechanically tested. Control samples consisted of hydrogels that were kept in PBS during the experiment or hydrogels that were kept in 250 mM CaCl_2_ during the assay. All conditions were assessed in triplicate.

### 2.6. Water Sorption Assay

Hydrogels were formed and immediately weighed (*m*_0_). Then, hydrogels were immersed in milliQ water for up to a week and weighed at 3, 24, 48, 72 and 168 h. The amount of water absorbed was calculated as follows:Water sorption (%) = [(*m*_t_ − *m*_0_)/*m*_0_] × 100(1)
where *m*_t_ is the weight of the hydrogel at a certain time and *m*_0_ is the weight of the hydrogel after formation. Hydrogels reached equilibrium after 24 h. Samples were prepared in triplicate.

### 2.7. Fibronectin Release

FN release at 24 h was quantified as a way to demonstrate that PEGylated FN was covalently crosslinked to the alginate network. To do so, hydrogels (2 wt% alginate with and without FN) were immersed for 24 h in PBS to assess the release of FN. Solutions from supernatants were used to quantify the amount of protein released by bicinchoninic acid colorimetric assay (MicroBCA assay kit, ThermoFisher Scientific, Eindhoven, The Netherlands), following the manufacturer’s instructions. Briefly, samples and standards were loaded onto a 96-well microplate and mixed with the working reagent (2 h, 37 °C). After the incubation, the microplate was let to cool down and the absorbance at 562 nm was measured using a plate reader (BIOTEK, Bad Friedrichshall, Germany).

### 2.8. Statistical Analysis

The statistical analysis was performed using GraphPad Prism 6.01 software (San Diego, CA, USA). All experiments were carried out in triplicate. All graphs represent mean ± standard deviation (SD). The goodness of fit of all datasets was assessed via D’Agostino–Pearson Normality test. When comparing three or more groups of normal distributed populations, differences were analysed by an analysis of variance test (ANOVA test) performing a Tukey’s post hoc test to correct for multiple comparisons. When populations were not normally distributed, differences were assessed by Kruskal–Wallis test with a Dunn’s post hoc test to correct for multiple comparisons. When comparing two groups, a t-test was performed. Differences among groups are shown as follows: for *p*-values <0.05—*, *p*-values <0.01—**, for *p*-values < 0.005—***, for *p*-values < 0.001—****, and if differences between groups are not statistically significant—n.s.

## 3. Results

### 3.1. Covalently Crosslinked Alginate Hydrogels with Tuneable Mechanical Properties that Incorporate Fibronectin and Protease-Degradable Cues

Alginate hydrogels with controlled mechanical properties were fabricated by utilising a norbornene-modified alginate that allows the covalent linkage of thiolated moieties via UV-initiated thiol-ene chemistry (Figure 1). The thiolated moieties randomly react with the neighbouring norbornene groups present on different alginate polymer chains. By using this strategy, alginate hydrogels can be prepared using different crosslinkers. For instance, non-degradable thiolated molecules such as dithiothreitol (DTT, Figure 1a) or degradable peptides that contain sequences cleavable by proteases (Figure 1b). Furthermore, thiol-ene chemistry allows the incorporation of other cell interactive cues such as the full protein fibronectin (FN, Figure 1c) resulting in an alginate-FN network. To incorporate FN, we functionalized it first with a poly(ethylene) glycol (PEG) polymer containing maleimide groups (four-arm-PEG-maleimide, Figure 1c). Maleimide groups spontaneously react with thiol groups at physiological pH via Michael-type addition reaction. This PEG addition process or PEGylation allowed FN to present the maleimide groups so it could be covalently crosslinked to the alginate network. The addition of protease-cleavable peptides and FN into the alginate network allows cells to directly interact with the hydrogel material.

In order to control the mechanical properties of the alginate hydrogels, we used different amounts of DTT crosslinker to achieve different crosslinking densities. To this end, we tested crosslinking densities from 0.05 to 1 (crosslinking ratios were calculated based on thiol to norbornene molar ratio, Table 1) (Figure 2). Figure 2a–d show the rheological behaviour of 2 wt% alginate (final concentration of alginate in the hydrogel was 20 mg·mL^−1^) hydrogels during a frequency sweep. Frequency sweeps were performed prior to performing strain sweeps to determine the regime where the storage modulus, G’, was independent of frequency. Figure 2e shows the elastic modulus measured from strain sweeps at an angular frequency of 10 rad·s^−1^ of 2 wt% alginate hydrogels with different crosslinking densities. The elastic modulus increases monotonically with the crosslinking ratio up to a maximum stiffness reached at crosslinking ratio of 0.5 and followed by a plateau at crosslinking ratio of 1. The crosslinking ratio is calculated as the ratio between thiol groups to norbornene groups and, thus, a crosslinking ratio of 0.5 means that for every thiol group in the system, there are two norbornene groups available (i.e., there are free norbornenes after crosslinking). A crosslinking ratio of 1, where theoretically all norbornene groups present in the alginate backbone have reacted with thiol groups from DTT, showed a similar value of stiffness compared to crosslinking ratio 0.5, where the maximum in stiffness is measured. This suggests that the system is already saturated at ratio 0.5 (i.e., where the maximum stiffness is observed) and that the addition of more crosslinker (e.g., X = 1) does not contribute to increasing the final bulk mechanical properties of the hydrogel. Using 2 wt% alginate allowed us to fabricate hydrogels with a maximum stiffness of 4.7 ± 0.2 kPa (X = 0.5). Figure 2f shows the elastic modulus of 3 wt% alginate hydrogels (final concentration of alginate in the hydrogel was 30 mg·mL^−1^) with different crosslinking ratios (0.05 to 1, Table 1) and measured from strain sweeps at an angular frequency of 10 rad·s^−1^ (Appendix A). Note that frequency sweeps were also performed for 3 wt% alginate hydrogels to determine the range where G’ was independent of frequency (data not shown). By increasing the percentage of alginate in the system—i.e., by using 3 wt% instead of 2 wt%—we sought to increase the number of norbornene groups available to react, which leads to higher crosslinking densities. Matching the results obtained from 2 wt% alginate hydrogels, the mechanical properties increased when increasing the crosslinking ratio up to a plateau. The maximum stiffness achieved with 3 wt% alginate hydrogels was 8.7 ± 0.2 kPa. Covalent crosslinking of alginate permits control over mechanical properties of the system. For hydrogels fabricated using 2 wt% alginate, the stiffness range varied from 0.2 ± 0.1 (X = 0.05) to 4.2 ± 0.8 kPa (X = 1), and for hydrogels containing 3 wt% alginate, stiffness varied from 0.4 ± 0.0 (X = 0.05) to 8.8 ± 0.8 kPa (X = 1). For the sake of simplicity, the remaining experiments were carried out using 2 wt% alginate hydrogels with crosslinking ratios of 0.1 or 0.5.

### 3.2. Alginate-Fibronectin Hydrogels with Tuneable Mechanical Properties and Enzymatic Degradation

Next, we incorporated degradable peptide crosslinkers (VPM) to induce susceptibility to enzymatic cleavage upon exposure to proteases. We also introduced full-length FN (VPM-FN), which contains cell-adhesion domains to foster direct cell–material interactions by integrin binding to the hydrogel backbone. Furthermore, it can also sequester various growth factors to promote the signalling cascade and enhance cell differentiation. Figure 3a,b show representative frequency sweeps for degradable 2 wt% alginate hydrogels, with an X = 0.5 crosslinking ratio, and with degradable crosslinker without FN (VPM) or with FN (VPM-FN). The frequency sweeps for both systems present a linear behaviour at low frequencies (<100 rad·s^−1^). When reaching higher frequencies, we observed that G’ increases for VPM hydrogels but decreases for VPM-FN hydrogels; however, the characterization of the hydrogels was carried out within the linear viscoelastic regime—that is to say, at lower frequencies. Figure 3c shows the elastic modulus measured from strain sweeps at an angular frequency of 10 rad·s^−1^, which differs for gels without (VPM) and with FN (VPM-FN), with values of 2.6 ± 0.3 and 5.1 ± 1.9 kPa, respectively. This increase in Young’s modulus for the VPM-FN hydrogels with the same amount of crosslinker (X = 0.5, Table 1) as VPM hydrogels without FN, is due to the fact that FN is covalently linked to the alginate and thus the crosslinking density in this system is higher and enough to observe differences in stiffness.

Figure 3d shows the percentage of equilibrium water sorption by degradable alginate hydrogels (X = 0.5) without (VPM) and with FN (VPM-FN). Water sorption was measured, recording weights before and after immersion in milliQ water (i.e., no dry weights were recorded). Both types of hydrogels absorb similar amounts of water and therefore the presence of FN does not have an effect on water diffusion through the system.

We also confirmed the covalent bonding of FN to the norbornene-modified alginate by measuring FN release from the hydrogels (Figure 3e). If FN is bound to alginate, the protein should be retained in the hydrogel (no diffusion). We compared alginate hydrogels with and without FN by immersing them in PBS for 24 h and collecting the supernatants to measure the amount of FN protein present in the supernatant via colorimetric assay. We observed no differences in protein release for alginate hydrogels (VPM) and alginate-FN hydrogels (VPM-FN). The absorbance values observed were low (around 0.4 a.u), meaning that no protein was detected.

### 3.3. Dual (Covalent and Ionic) Crosslinked Alginate Hydrogels with Dynamic and Reversible Control over Mechanical Properties

We took advantage of the dual ionic and covalent crosslinking that norbornene-modified alginate offers (Figure 4) to investigate the potential of this system to permit dynamic and reversible control over stiffness (Figure 5 and Figure 6). We first polymerised 2 wt% alginate hydrogels using DTT as a crosslinker and UV light (covalent crosslinking), with crosslinking ratios of X = 0.1 and X = 0.5 (low and high initial crosslinking ratios, respectively). After covalent polymerisation, the mechanical properties of the hydrogels were measured, obtaining similar values to those obtained previously (Figure 2e). Then, we immersed the hydrogels in a calcium bath for 2 h (ionic crosslinking) and measured the mechanical properties again (Figure 5a). Ionic crosslinking was accompanied by a significant shrinkage of the hydrogel, which is shown in Figure 5d,e and Figure 6d,e. This observation correlates with diameter measurements of the hydrogels (Figure 5f,g and Figure 6f,g). The stiffness of the hydrogels following the calcium crosslinking was significantly higher than the initial stiffness measured after covalent crosslinking. For hydrogels with X = 0.1, the Young’s modulus increased from 2.8 ± 1.0 to 50.9 ± 12.5 kPa (Figure 5b and Figure 6b) and for X = 0.5 hydrogels, the Young’s modulus increased from 5.2 ± 1.0 up to 86.6 ± 29.6 kPa (Figure 5c and Figure 6c), which is a 19- and 17-fold increase, respectively (Table 2).

After ionic crosslinking, we sought to investigate whether or not the calcium crosslinking could be effectively reversed or quenched. We tested the quenching effect of a PBS bath (PBS without Ca^2+^ or Mg^2+^) to passively remove the calcium present in the system (Figure 5). We also investigated the effect of EDTA, a chelating agent that binds divalent cations to actively sequester the calcium present in the hydrogel (Figure 6).

We kept the hydrogels immersed in either PBS or EDTA for 3 days and then measured the mechanical properties of the hydrogels. Overall, the mechanical properties drastically dropped, with some differences (Figure 5b,c and Figure 6b,c). Hydrogels that were kept in PBS softened down to 7.3 ± 1.8 and 8.4 ± 0.1 kPa (for X = 0.1 and 0.5, respectively) whereas hydrogels immersed in EDTA softened to initial stiffness values (2.3 ± 2.7 kPa for X = 0.1 and 5.9 ± 0.1 kPa for X = 0.5). Note that stiffnesses measured after UV crosslinking and after quenching in EDTA are not statistically different.

Finally, after calcium quenching, we immersed the hydrogels in another calcium bath for 2 h (second ionic crosslinking) and measured the mechanical properties. Overall, the hydrogels stiffened up to similar values obtained in the first immersion in calcium solution (Table 2). Note that the mechanical properties of hydrogels after calcium treatment are not statistically different. Hydrogels quenched with PBS recovered stiffness values of the first ionic crosslinking in both cases (X = 0.1 and 0.5, Figure 5b,c). Interestingly, hydrogels that were quenched with EDTA showed different results (Figure 6b,c). For hydrogels with low crosslinking (X = 0.1), stiffening of the hydrogel was below the one measured in the first cycle of stiffening (first ionic crosslinking). For hydrogels with high crosslinking ratio (X = 0.5), the stiffness values were higher than the ones measured in the first cycle of stiffening. This indicates that the initial covalent crosslinking could be critical for maintaining several stiffening–softening cycles.

## 4. Discussion

Alginate hydrogels are blank slate scaffolds used in biomedical applications such as drug delivery [18,52], wound dressing [53] or tissue engineering [54]. In this study, we used a modified alginate that incorporates the functional group norbornene, which can react in a UV-initiated thiol-ene polymerisation. Alginate hydrogels were formed by covalent crosslinking using thiolated crosslinkers such as DTT or custom-synthesised peptides (Figure 1). Thiol-ene chemistry has been widely used for hydrogel photocrosslinking and it is cytocompatible [29,33,35]. Covalent crosslinking chemistries allowed us to control the mechanical properties of the hydrogels (Figure 2); by varying the crosslinking ratio X, we could vary the crosslinking density of the system and therefore control the final bulk properties of the material. This has been shown before for other hydrogel systems fabricated by covalent crosslinking [5,55,56].

By using thiol-ene chemistry, we were able to incorporate protease-degradable peptides as crosslinkers (VPM, Figure 3). The use of these custom-synthesised and protease-sensitive peptides facilitates cell migration and spreading within hydrogels, which can also direct stem cell differentiation [56] and promote cell infiltration in vivo [29]. This strategy is also important in drug delivery systems [19,57,58]. Furthermore, we could also incorporate the full-length FN, an important extracellular matrix protein. The strategy used to covalently link FN to the norbornene-modified alginate involved first modifying FN itself via PEGylation [47,48]. PEGylation consists of grafting a few molecules of PEG (in this case a four-arm-PEG-maleimide) so FN is functionalised with maleimide groups that spontaneously react with thiol groups via Michael-type addition reaction at physiological pH (Figure 1c). By utilizing a spontaneous reaction (thiol-maleimide) and taking into account that FN contains several thiol groups in its structure, the final number of available maleimide groups will range up to a maximum of 12 available groups (as each FN molecule is reacted with four molecules of four-arm-PEG-maleimide). This strategy has been previously used to incorporate FN in other hydrogel systems [47] as well as other proteins such as fibrinogen [48]. Figure 3 demonstrates the covalent incorporation of FN into the hydrogel. Similar results were previously obtained for FN-PEG and FN-hyaluronic acid hydrogels [47,50]. We also characterised the swelling behaviour of the VPM and VPM-FN hydrogels since diffusion is critical when using hydrogels as carriers to release therapeutic or biologically active molecules. Alginate gels have been widely studied for the delivery drugs of different molecular weights. This is because alginate hydrogels are usually nanoporous (with a mesh size of approximately 5 nm), which leads to rapid diffusion of small molecules. For instance, flurbiprofen was released in less than 2 h from alginate gels ionically crosslinked [52]. To regulate the kinetics of drug release, alginate hydrogels have shown to be most useful when there is a bond between the drug and the alginate. For example, the combination of ionic and covalent crosslinking of alginate led to a more sustained release of flurbiprofen due to the increase in crosslinking and subsequent decrease in swelling [57].

The presence of FN in the hydrogel maintains the ability to hold water with increased mechanical properties. By adding FN (VPM-FN), which can only react with the crosslinker VPM, the network of the hydrogel changes. This change in the network increases the number of elastically active network chains. Therefore, it is reasonable to think that the mechanical properties of this network will change as well. In addition, the FN molecule adds rigidity to the network as it presents a high molecular weight (~220 kDa per monomer unit). The change in the number of elastically active network chains together with the presence of FN contribute to the increase in the final bulk stiffness observed for the VPM-FN hydrogels. FN is also a hydrophilic molecule [59], and the use of PEGylated FN that contains a few molecules of PEG (another hydrophilic molecule) allows for additional water in the system, which could explain VPM-FN water uptake being similar to VPM hydrogels, despite the differences in bulk stiffness. This could also improve the efficiency of delivery of cargo molecules such as growth factors that have a binding site present on FN [48,60,61].

Finally, alginate hydrogels can be also fabricated by ionic crosslinking using divalent cations and therefore we investigated the mechanical reversibility of dual crosslinking (covalent and ionic, Figure 5 and Figure 6). As a proof of concept, we chose to immerse the hydrogels for 2 h in a 250 mM CaCl_2_ bath. The time for ionic gelation was chosen to allow high ionic crosslinking. We note that time is a critical parameter for gelation but also for cytocompatibility. Gelation rates also control gel uniformity and strength. Normally, slower gelation rates produce more uniform structures and greater mechanical integrity [21]. Therefore, ionic gelation kinetics have to be carefully selected depending on the final application. Gelation can be slowed down by the use of calcium sources with lower solubility in water such as calcium sulphate (CaSO_4_) and calcium carbonate (CaCO_3_) [21]. Moreover, alginate can be ionically crosslinked with other divalent cations such as Mg^2+^, Ba^2+^, Mn^2+^, Sr^2+^ or Zn^2+^ [62,63,64]. Each divalent cation presents a different affinity to alginate, which also depends on the M/G residue ratio of alginate. Further, trivalent cations of iron (Fe^3+^) have been also described to bind alginate [65]. Consequently, ionic gelation can be finely tuned by the use of different cations that bind alginate. The use of different divalent cations could also trigger different biological outputs. For instance, alginate hydrogels with dual crosslinking (covalent + ionic) using Mg^2+^ showed improved osteoblast attachment [66]. Alginate hydrogels ionically crosslinked with up to 500 mM CaCl_2_ have been reported to be cytocompatible [67], suggesting that the concentration of calcium chosen in our work is compatible with cell studies.

We also sought to investigate whether or not the observed stiffening of hydrogels due to the effect of ionic crosslinking could be neutralised by quenching the calcium ions. To do so, we used two different quenching methods, PBS and EDTA (Figure 5 and Figure 6). Both methods were able to soften the gels to values similar to the ones measured after covalent crosslinking (Table 2). PBS acts as a passive quencher, as phosphates compete with the carboxylate groups of alginate for binding calcium [21]. On the other hand, EDTA acts as an active quencher, as it is a calcium chelator. We observed some differences in the mechanical properties comparing the use of PBS and EDTA. EDTA quenching method resulted in stiffness values closer to initial values (after covalent crosslinking, not statistically different), whereas PBS resulted in similar stiffness values but these were statistically different. This result suggests that EDTA is more effective at removing calcium from the hydrogel compared to PBS, which was as expected since EDTA actively binds calcium. For this proof of concept experiment, we used 0.25 M of EDTA, but this result opens up the possibility to explore other concentrations of EDTA to optimise softening kinetics or even the use of other known calcium chelators such as sodium citrate [20]. On the other hand, the use of PBS as a quencher is advantageous in terms of cytocompatibility if these gels are loaded with cells. After calcium quenching, we showed that stiffening is possible by immersing the gels in the CaCl_2_ bath again (and therefore crosslinking the gels ionically again). This indicates that the softening step is reversible and that these hydrogels could undergo a number of stiffening–softening cycles (note that we only tested one cycle of stiffening–softening). 

The ability to fabricate a hydrogel system with dynamic, tuneable and reversible mechanical properties, which can incorporate full proteins such as FN and, furthermore, be degraded on-demand by proteases, is of relevance in the biomaterial field. In addition, the conditions tested have been shown to be compatible with cell encapsulation, which allows the use of these dual crosslinked alginate hydrogels as 3D microenvironments to model physiological processes with dynamic changes in stiffness, such as fibrosis.

## 5. Conclusions

We developed alginate-fibronectin hydrogels that, taking advantage of dual covalent and ionic crosslinking, allow for dynamic and reversible control of mechanical properties. Our strategy takes advantage of a norbornene-modified alginate, which can be used in mild reactions to covalently bind different thiolated molecules. We demonstrated covalent incorporation of cues that cells can interact with, such as degradable crosslinkers and full-length proteins such as FN. As the thiol-ene chemistry used here was known to be cytocompatible, this system could find applications in 2D and 3D cell cultures in vitro. Furthermore, we took advantage of dual covalent and ionic crosslinking using alginate to demonstrate that the mechanical properties of this system were tuneable and could undergo cycles of stiffening–softening in a dynamic, tuneable and reversible way. The amounts of CaCl_2_ used are compatible with cell culture. Furthermore, the stiffening–softening kinetics could be further adjusted by optimising ionic crosslinking methods—i.e., using different divalent cations, optimising the source and amount of divalent cations or adjusting quenching methods. Such a versatile hydrogel system with dynamic and reversible control of mechanical properties, incorporating biologically interactive features such as degradability and full-length proteins, could offer multiple opportunities for biomedical applications such as drug delivery, wound dressings or tissue engineering.

## Figures and Tables

**Figure 1 polymers-13-00433-f001:**
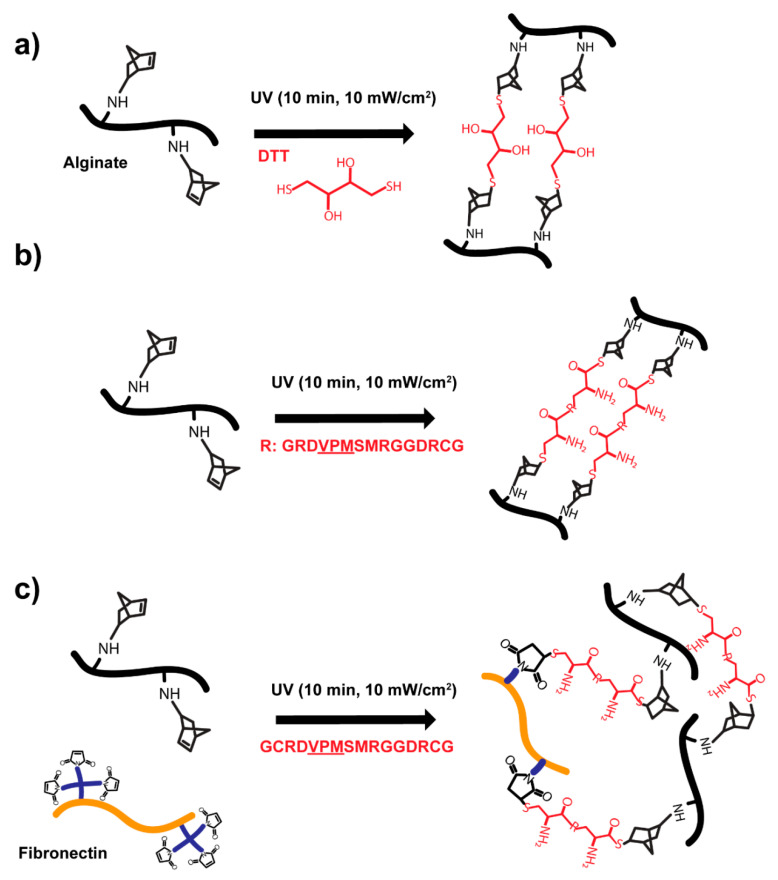
Simplified sketch of covalently crosslinked alginate hydrogels with degradable or non-degradable crosslinks. (**a**) Norbornene-modified alginate hydrogels are formed by UV-initiated polymerisation using a thiolated crosslinker (dithithreitol, DTT, in red) or (**b**) a custom-synthesised thiolated protease-sensitive peptide (GCRDVPMSMRGGDRCG, VPM, in red, where R depicts the degradable peptide chain). (**c**) Additionally, alginate hydrogels can incorporate the full fibronectin protein by using maleimide functionalised PEGylated fibronectin (PEGylation using a 4-arm-PEG-maleimide, ratio 1:4 FN:PEGMAL) and a thiolated protease-sensitive peptide crosslinker. Note that both norbornene-thiol and maleimide-thiol reactions are stochastic and the sketches aim to represent the nature of the different crosslinking reactions, not the real final structure of the hydrogel.

**Figure 2 polymers-13-00433-f002:**
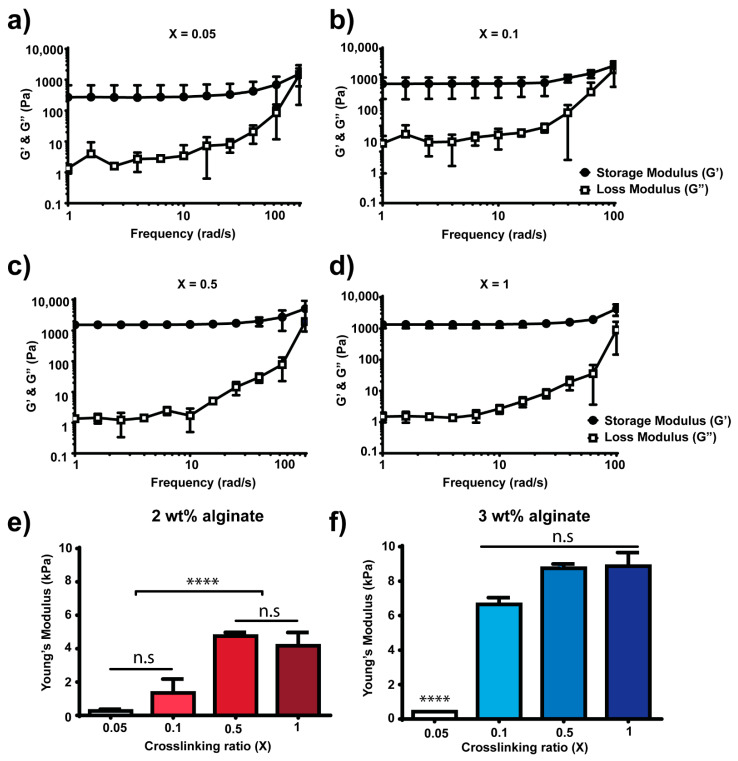
Mechanical properties of non-degradable DTT crosslinked alginate hydrogels; 2 wt% and 3 wt% alginate hydrogels were polymerised with different thiol (from DTT crosslinker) to norbornene (from alginate) crosslinking ratios (X = 0.05 to 1). (**a**–**d**) Representative frequency sweep curves depicting the storage (G’) and loss (G”) moduli of different 2 wt% alginate hydrogels polymerised with different X ratios (Table 1). Young’s modulus obtained from strain sweep curves, at frequency of 10 rad·s^−1^ and at different X ratios for 2 wt% (**e**) and 3 wt% (**f**) alginate hydrogels, respectively. **** *p*-values < 0.001, n.s. not statistically significant.

**Figure 3 polymers-13-00433-f003:**
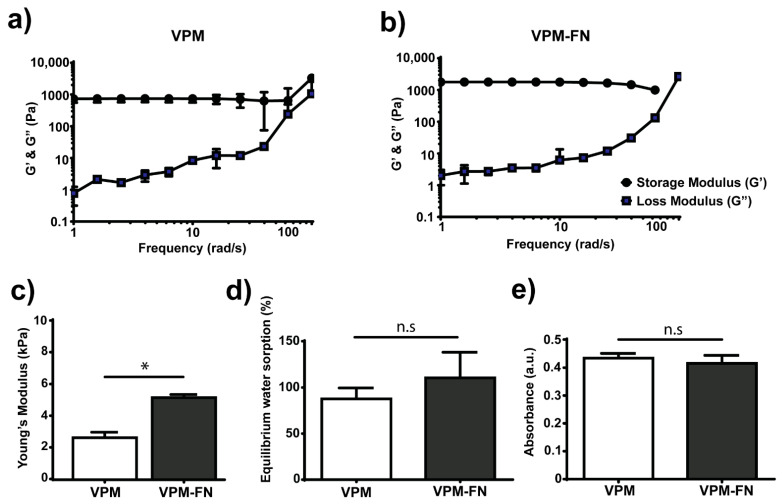
Enzymatically degradable alginate hydrogels incorporating the full-length fibronectin (FN) protein and characterized for mechanical, swelling and protein release properties. Frequency sweep of 2 wt% alginate-degradable hydrogels covalently crosslinked with the enzymatically degradable crosslinker VPM (X = 0.5) (**a**) without (VPM) and (**b**) with additional full-length FN protein (VPM-FN). (**c**) Young’s modulus calculated from strain sweep experiments with or without FN (VPM or VPM-FN). (**d**) Percentage of water uptake at equilibrium (24 h) of hydrogels with or without FN (VPM or VPM-FN). (**e**) Protein amount detected in supernatant of hydrogels with or without FN (VPM or VPM-FN) (arbitrary units (a.u.)). * *p*-values < 0.05, n.s. not statistically significant.

**Figure 4 polymers-13-00433-f004:**
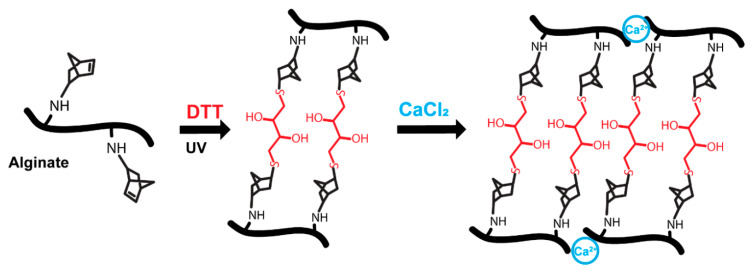
Sketch of dual crosslinking approach combining covalent and ionic crosslinking. Hydrogels were covalently crosslinked via photopolymerisation with DTT and then immersed in a solution containing calcium (250 mM CaCl_2_) to allow for ionic crosslinking.

**Figure 5 polymers-13-00433-f005:**
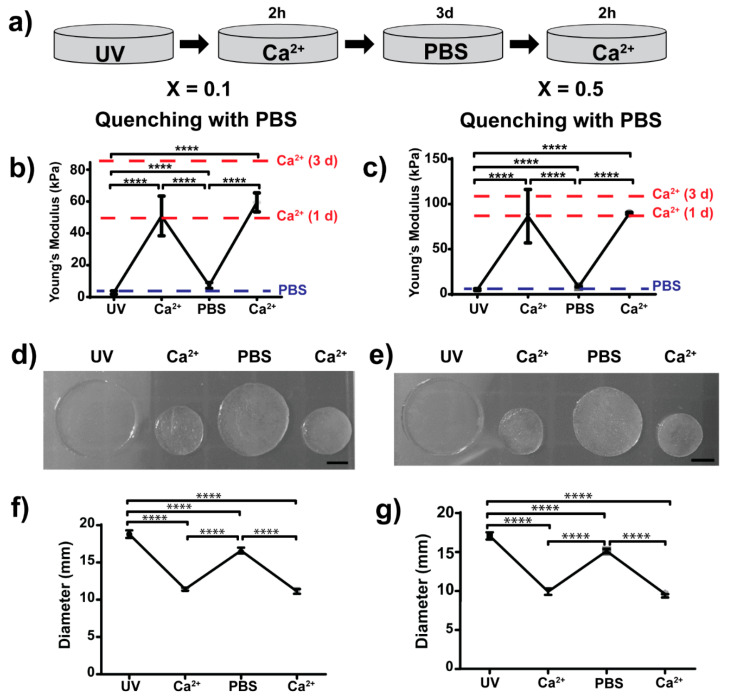
Dynamic and reversible control over mechanical properties and swelling behaviour of covalently crosslinked alginate hydrogels with low and high crosslinking ratios (X = 0.1 and X = 0.5) followed by additional reversible ionic crosslinking using calcium and quenching with phosphate buffer saline (PBS). (**a**) Procedure followed during a cycle of incorporation/removal of calcium to control mechanical properties, where first 2 wt% alginate hydrogels were polymerised by UV irradiation, then calcium is added for 2 h to allow ionic crosslinking. After that, calcium was removed by immersing in PBS for 3 days and, finally, hydrogels were ionically crosslinked again by addition of calcium for 2 h. Stiffness of (**b**) low crosslinked hydrogels (X = 0.1) and (**c**) highly crosslinked hydrogels (X = 0.5) was measured at each step of the dual crosslinking cycle using PBS as a quenching method. Controls of hydrogels measured in PBS or calcium solution are shown as dashed lines (marked as PBS (blue) or Ca^2+^ (red), respectively). (**d**,**e**) Representative images of hydrogels with low (X = 0.1) and high (X = 0.5) crosslinking, respectively (scale bar = 5 mm), and (**f**,**g**) hydrogel diameters measured at every step of the dual crosslinking cycle using PBS as a quenching method. **** *p*-values < 0.001.

**Figure 6 polymers-13-00433-f006:**
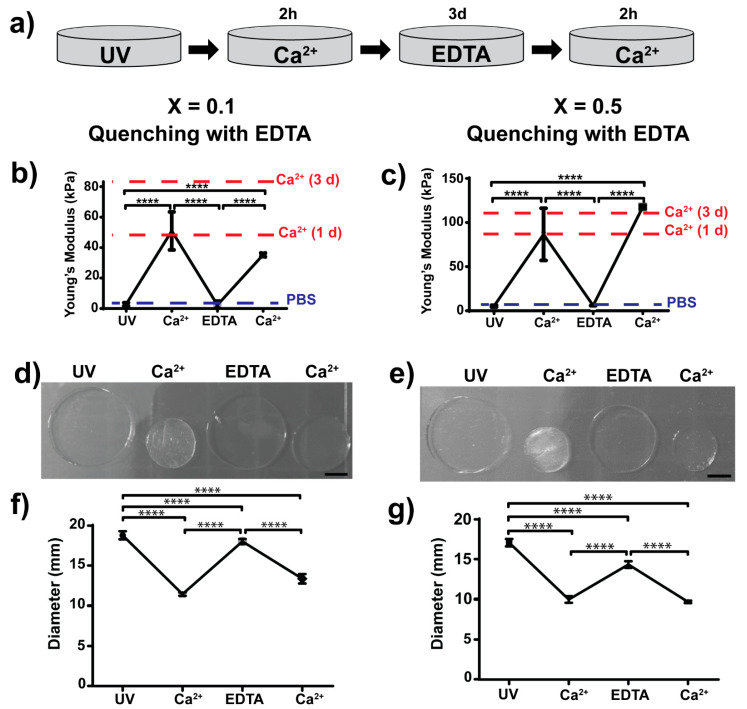
Dynamic and reversible control over mechanical properties and swelling behaviour of covalently crosslinked alginate hydrogels with low and high crosslinking ratios (X = 0.1 and X = 0.5) followed by additional reversible ionic crosslinking using calcium and quenching with EDTA. (**a**) Procedure followed during a cycle of incorporation/removal of calcium to control mechanical properties, where first 2 wt% alginate hydrogels were polymerised by UV irradiation, then calcium was added for 2 h to allow ionic crosslinking. After that, calcium was removed by immersion EDTA for 3 days and, finally, hydrogels were ionically crosslinked again by addition of calcium for 2 h. Stiffness of (**b**) low crosslinked hydrogels (X = 0.1) and (**c**) highly crosslinked hydrogels (X = 0.5) was measured at each step of the dual crosslinking cycle using EDTA as a quenching method. Controls of hydrogels measured in PBS or calcium solution are shown as dashed lines (marked as PBS (blue) or Ca^2+^ (red), respectively). (**d**,**e**) Representative images of hydrogels with low (X = 0.1) and high (X = 0.5) crosslinking (scale bar: 5 mm), and (**f**,**g**) hydrogel diameters measured at every step of the dual crosslinking cycle using EDTA as a quenching method. **** *p*-values < 0.001.

**Table 1 polymers-13-00433-t001:** Composition of alginate hydrogels.

2 wt% Alginate	Alginate (mg·mL^−1^)	Crosslinking Ratio X (Thiol to Norbornene Groups)	DTT (µg·mL^−1^)	
	20	0.05	30	
	20	0.1	60	
	20	0.5	305	
	20	1	610	
**2 wt% Alginate**	**Alginate** **(mg·mL^−1^)**	**Crosslinking Ratio X** **(Thiol to Norbornene Groups)**	**VPM** **(µg·mL^−1^)**	**FN** **(µg·mL^−1^)**
	20	0.5	3378	1000
**3 wt% Alginate**	**Alginate** **(mg·mL^−1^)**	**Crosslinking Ratio X** **(Thiol to Norbornene Groups)**	**DTT** **(µg·mL^−1^)**	
	30	0.05	46	
	30	0.1	92	
	30	0.5	460	
	30	1	920	

**Table 2 polymers-13-00433-t002:** Young’s modulus (mean ± SD) of covalently crosslinked alginate hydrogels measured during stiffening–softening dual crosslinking.

2 wt% Alginate	Young’s Modulus (kPa)	2 wt% Alginate	Young’s Modulus (kPa)
X = 0.1	Mean ± SD	X = 0.5	Mean ± SD
Covalent crosslinking	2.8 ± 1.0	Covalent crosslinking	5.2 ± 1.0
Ionic crosslinking	50.9 ± 12.5	Ionic crosslinking	86.6 ± 29.6
Calcium quenching (PBS)	7.3 ± 1.8	Calcium quenching (PBS)	8.4 ± 0.1
Ionic crosslinking	59.5 ± 5.9	Ionic crosslinking	89.6 ± 1.1
X = 0.1	Mean ± SD	X = 0.5	Mean ± SD
Covalent crosslinking	2.8 ± 1.0	Covalent crosslinking	5.2 ± 1.0
Ionic crosslinking	50.9 ± 12.5	Ionic crosslinking	86.6 ± 29.6
Calcium quenching (EDTA)	2.3 ± 2.7	Calcium quenching (EDTA)	5.9 ± 0.1
Ionic crosslinking	35.2 ± 1.1	Ionic crosslinking	117.6 ± 1.7

## Data Availability

Data available in a publicly accessible repository https://dx.doi.org/10.17617/3.5k.

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
