# Peer review of "Dynamic Mechanical Control of Alginate-Fibronectin Hydrogels with Dual Crosslinking: Covalent and Ionic"

_polymers, 2021, doi:10.3390/polym13030433_

Round 1

Reviewer 1 Report

The paper is called “Dynamic mechanical control of...hydrogels with dual covalent and ionic crosslinking”

What is exactly meant by “dynamic mechanical control”? I think that the control of gel properties is realized through change of crosslinker concentration in all cases including immersion of gels into the ion-containing solutions. I consider the title and the term dynamic control as well as reversible control nonsense.

Also, I think that the title should be reformulated: at present form, it introduces dual covalent crosslinking – and one ionic crosslinking, does not it? I propose for example this correction:

“...hydrogels with dual crosslinking: covalent and ionic”

The fact, that the gel properties change with crosslinker (both chemical and physical) concentration is not new. Quite the opposite: the crosslinker effect (concentration, functionality, reaction type) is the long-lasting fundament in the area. However, the experimental system presented here is interesting and could be published if the background and novelty of this work are presented clearly. But I am convinced that it is important to use appropriate language and correct schemes to familiarize reader with the concept of the work correctly – see the comments below.

Add Abstract and farther: Degradability of a material is not a biological feature.

Add Introduction, Conclusion: what is the specific advantage of these gels/or changes of their stiffness for drug delivery? Experiments giving some context are missing, pls explain and give references.

Part 2.2. Preparation of PEGylated FN: was the given molar ratio FN : PEGMAL = 1 : 4 related to molecules or functional groups? How many of the maleimide  groups were reacted per one 4-arm PEG unit? In this connection – pls include the structure of four-functional PEGMAL in the scheme on Figure 1  -  otherwise the linear orange chain does not reflect the real structure.

Part 2.4.

The given relation between E and G’ is wrong. It reads: E =2G(1+n), so E = 3G’ if the Poisson ratio is close or equal to ½.  Was the data calculated according to the wrong formula or is it just a typing mistake?

How the value of strain for frequency measurements was chosen? Were the amplitude sweeps performed to find the linear viscoelastic region?

Part 3

Explain why the crosslinking ration thiol/norbornene 0.5 should provide all norbornene groups reacted?  - cf. Part 3, line 195-196. Pls specify precisely how the ratio in the Table 1 is defined: it should be thiol groups vs. norbornene groups, not thiol-containing units vs. norbornene group.

In this connection: the scheme of crosslinked chains on Figure 1 is misleading – it is not very likely to obtain such ladder structures, most likely, the neighboring crosslinked groups on one chain will connect with other two different chains through the space.    

Pls specify how many binding sites the FN has.

What is the difference between G’, G’’ determined from frequency sweep at 10 rad/s and the G’, G’’ determined from strain sweep at 10 rad/s? If the values are not equal, it means that the frequency sweep is measured at non-linear viscoelastic region. The strain sweep should be done first to find the linear viscoelastic behavior (LVR)– then the frequency sweep at correct strain, i.e. within LVR. Pls revise appropriate parts.  

There is no discussion related to course of G’, G’’ on frequency: e.g. the reasons for increase of G’ with frequency and in one case – the gel VPM-FN decrease of G’ with frequency.

What is the reason for decreasing E with crosslinking ratio  (Fig. 1, e/)?

The discussion on increase of crosslink density in VPM-FN gels is not convincing and not complete. Crosslink density should be referred to number of network chains per volume of network – because larger swelling for VPM-FN gel, the apparent crosslink density would decrease vompared with VPN-gel but because of adding four-functional PEGMAL, the number of elastically active network chains increases compensating and even overacting the effect of higher swelling.   

Part 4

First sentence: ...alginate hydrogels are used ...in drug delivery, wound dressing, tissue engineering – pls give references of use of alginate hydrogels in the applications.

Explain how the equilibrium swelling values given in this paper are related to diffusion in the gels: reconcile discussion between the lines 355-359. Diffusion should be linked with kinetics of swelling. Explain how is the swelling of gels in aqueous environment related to their release capacity – cf. line 356.

Line 358: if the crosslink density increases, swelling should decrease unless the system changed composition which is the case here – adding FN means adding GEG moieties, too. The crosslink density parameter is proportional to equilibrium modulus – such property was not determined in this work so the discussion is not well based.

Line 385 phosphates are anions, so they can compete with other anions – so with carboxylates, is that what you wanted to say?

Reviewer 2 Report

Well written and thought out paper. 

some comments: 

line 151 water sorption. 'immeadately weighed and immersed'. no dry weights were recorded which make calculations of WU, EWC etc difficult. 

line 195: all backbone reacted. where is the evidence for this? 

Figures 5 and 6 are similar and I suggest combining these. 
